# Synthesis of Novel Kaolin-Supported g-C_3_N_4_/CeO_2_ Composites with Enhanced Photocatalytic Removal of Ciprofloxacin

**DOI:** 10.3390/ma13173811

**Published:** 2020-08-28

**Authors:** Zhiquan Huang, Leicheng Li, Zhiping Li, Huan Li, Jiaqi Wu

**Affiliations:** 1College of Geosciences and Engineering, North China University of Water Resources and Electric Power, Zhengzhou 450046, Henan, China; huangzhiquan@ncwu.edu.cn (Z.H.); lizhiping@ncwu.edu.cn (Z.L.); lihuan@ncwu.edu.cn (H.L.); wujqwangyi@163.com (J.W.); 2Luoyang Institute of Science and Technology, Luoyang 471023, Henan, China

**Keywords:** kaolin, CeO_2_, g-C_3_N_4_, visible-light irradiation, photocatalytic degradation

## Abstract

Herein, novel ternary kaolin/CeO_2_/g-C_3_N_4_ composite was prepared by sol-gel method followed by hydrothermal treatment. The self-assembled 3D “sandwich” structure consisting of kaolin, CeO_2_ and g-C_3_N_4_ nanosheets, was systematically characterized by appropriate techniques to assess its physicochemical properties. In the prerequisite of visible-light irradiation, the removal efficiency of ciprofloxacin (CIP) over the kaolin/CeO_2_/g-C_3_N_4_ composite was about 90% within 150 min, 2-folds higher than those of pristine CeO_2_ and g-C_3_N_4_. The enhanced photocatalytic activity was attributed to the improved photo-induced charge separation efficiency and the large specific surface area, which was determined by electrochemical measurements and N_2_ physisorption methods, respectively. The synergistic effect between the kaolin and CeO_2_/g-C_3_N_4_ heterostructure improved the photocatalytic performance of the final solid. The trapping and electron paramagnetic resonance (EPR) experiments demonstrated that the hole (h^+^) and superoxide radicals (•O_2_^−^) played an important role in the photocatalytic process. The photocatalytic mechanism for CIP degradation was also proposed based on experimental results. The obtained results revealed that the kaolin/CeO_2_/g-C_3_N_4_ composite is a promising solid catalyst for environmental remediation.

## 1. Introduction

Water pollutants have attracted considerable attention from all levels of society because of the increasing challenges in environmental concerns and energy demands [1]. Harmful pollutants, such as antibiotics, dyes and heavy metals, pollute the water significantly and threaten human health. Ciprofloxacin (CIP), a commonly used antibiotic in all aspects of life [2], can have a serious impact on the ecological environment balance [3]. To date, several strategies, including electro-Fenton [4], biological degradation [5], adsorption [6], membrane filtration [7] and photocatalytic degradation [8], have been employed to remove contaminants. Among them, photocatalysis has attracted more attention for the degradation of antibiotics [9]. Due to the cost-effectiveness and excellent stability, photocatalysts are considered ideal solids for water purification [10]. However, traditional semiconductor-based photocatalysts exhibit poor photoactivity due to the low utilization of solar energy and rapid recombination of photo-induced electrons and holes, consequently, their application in the removal of organic pollutants is restricted [11,12].

Graphitic carbon nitride (g-C_3_N_4_) has received great attention due to its unique electronic structure, high thermal and chemical stability, high hardness, low density, good biological compatibility and high wear resistance [13,14,15]. However, poor solar energy utilization, low specific surface area and low number of active sites [16] are the main factors explaining the low photocatalytic performance of g-C_3_N_4_ [17]. To improve the photocatalytic performance of g-C_3_N_4_, great effort has been made to modify the structure and texture by means of various strategies, such as elements doping [18,19], creation of porosity [20,21], noble metals deposition [21,22], loading metal nanoparticles [23,24] and construction of semiconductor heterojunctions [25,26,27]. Among numerous experimental procedures, semiconductor coupling has proven an excellent approach for promoting charge separation and photocatalytic performance [28]. One of the most important semiconductor materials is cerium oxide (CeO_2_), which is used in a wide range of industrial applications, such as catalytic reactions, solar and fuel cells. This is due to its appealing low cost, high chemical stability, high oxygen storage capacity and strong advantages [29,30,31,32,33]. In addition, CeO_2_ is an excellent photocatalyst due to the narrow band gap, and absorption of visible light [34,35]. Therefore, CeO_2_/semiconductor heterojunction formation not only improves conversion rates of light energy, but can accelerate the separation of electrons and holes, with a positive impact on the photocatalytic activity [36].

Kaolin, a nonmetal phyllosilicate clay mineral, is mainly composed of layers of silica tetrahedra and alumina octahedra, with good hydrophilic and adsorption properties [37,38,39,40]. Due to the large specific surface area and pore volume, it exhibits high adsorption capacity of organic compounds. As a result, it is often used as a carrier for photocatalytic and adsorption materials. Over the last few decades, kaolin/TiO_2_, kaolin/CeO_2,_ CeO_2_/g-C_3_N_4_ [41,42] and kaolin/g-C_3_N_4_ composites have been investigated [43,44,45]. However, the construction of ternary kaolin/CeO_2_/g-C_3_N_4_ heterojunction photocatalyst is yet to be reported. Due to the fact that kaolin has unique structural and adsorption properties, the combined photocatalyst should further improve the light utilization and accelerate the interfacial charge transfer. Therefore, it would be an excellent strategy to combine the g-C_3_N_4_/CeO_2_ heterostructure with a natural mineral, such as kaolinite, to synthesize a ternary photocatalyst with enhanced photocatalytic activity, since both adsorption capacity and adsorption of visible light can be increased [29,46,47].

Herein, a novel ternary kaolin/CeO_2_/g-C_3_N_4_ heterostructure was fabricated via a facile sol-gel method followed by a mild hydrothermal treatment. The structure and morphology of kaolin/CeO_2_/g-C_3_N_4_ heterostructure were characterized systematically by TEM, XRD, EDS, N_2_ Physisorption, XPS, Brunauer-Emmett-Teller (BET) theory, SEM, electrochemical impedance spectroscopy (EIS), EPR and UV-Vis diffuse reflectance spectra (DRS). The synthesized kaolin/CeO_2_/g-C_3_N_4_ heterostructure exhibited outstanding photocatalytic activity towards CIP under visible-light irradiation. Hence, 90% CIP degradation efficiency was obtained within 150 min, which was 2-folds higher than that of pristine g-C_3_N_4_. The reasons for enhanced photocatalytic activity originated from improved light harvesting and enhanced separation and transfer efficiency. Based on the obtained experimental results, the mechanism of photocatalytic degradation over kaolin/CeO_2_/g-C_3_N_4_ composite was also proposed.

## 2. Experimental

### 2.1. Materials

The purified kaolin was obtained from Jingdezhen, in the eastern province of Jiangxi province, China. NaOH, Ce(NO_3_)_3_•6H_2_O, ciprofloxacin, melamine, urea, ethanol and nitric acid were purchased from Aladdin industries and were of analytical grade.

### 2.2. Preparation of g-C_3_N_4_

The g-C_3_N_4_ powder was prepared according to previous work [48]. In a typical procedure, urea (15 g) and dicyandiamide (15 g) were mixed into a covered alumina crucible and heated at 550 °C (heating rate of 5 °C/min) for 4 h. The resulting yellow solid powder was collected for further use.

### 2.3. Preparation of Kaolin/CeO_2_ Composite

Ce(NO_3_)_3_•6H_2_O (1.305 g) and NaOH (14.400 g) were dissolved in distilled water (60 mL) under magnetic stirring for 2 h. Then, the homogeneous solution was transferred to a 100 mL Teflon-lined stainless steels autoclave and heated at 180 °C for 24 h. Subsequently, the autoclave was naturally cooled to room temperature. The resulting precipitate was washed three times with distilled water and ethanol. The product was added to deionized water (60 mL) and magnetically stirred for 2 h producing suspended solution A. Kaolin (1.0 g) was immersed in ethanol (30 mL), magnetically stirred for 30 min and sonicated for 2 h producing solution B. Then, suspended solution A was slowly added to suspension B with stirring for 12 h. The obtained product kaolin/CeO_2_ was dried in an oven at 80 °C for 12 h and heated at 450 °C (heating rate of 5 °C/min) for 3 h in a muffle furnace. For comparison, CeO_2_ was synthesized using a similar method without kaolin.

### 2.4. Preparation of Kaolin/CeO_2_/g-C_3_N_4_ Composite

A certain amount of the kaolin/CeO_2_ composite was dispersed in ethanol solution and stirred for 12 h, (solution A). Separately, a certain amount of g-C_3_N_4_ powder was dispersed in ethanol solution and stirred for at least 8 h, (solution B). Then, solution B was added dropwise to solution A and stirred continuously for 1 h, (solution C). The solution C was continuously stirred for 24 h at room temperature. The obtained suspension was centrifuged and the obtained solid dried in an oven at 60 °C for 12 h. The dried sample was ground and heated to 550 °C (heating rate of 5 °C/min) for 4 h in a muffle furnace. The obtained sample was a kaolin/CeO_2_/g-C_3_N_4_ composite.

### 2.5. Characterization

X-ray diffraction (XRD) was performed on a Bruker D8 Advance (Cu-Kα source, λ = 1.5418 Å, 45 KV, and 40 mA). The samples were scanned within the range of 5° to 80° 2θ with a scanning speed of 7°/min. X-ray photoelectron spectroscopy (XPS, Al Ka, hv = 1486.7 eV, reference C 1 s binding energy of 284.8 eV) was used to analyze the elemental composition and chemical state at the sample surface. A Lambd950 spectrophotometer (Perkin Elmer, Costa Mesa, CA, USA) was used to record the UV-Vis spectra between 200 and 800 nm. UV–Vis diffuse reflectance spectra were registered on a Hitachi-3900H spectrophotometer (Tokyo, Japan). SEM images were recorded on a S-4800 scanning electron microscope equipped with a 5.0 KV energy dispersive spectrometer produced by Hitachi high-tech Co. Ltd. (Tokyo, Japan) TEM and HRTEM images were collected on a field emission transmission electron microscope and JEM-3010 electron microscope, respectively, produced by JEOL (Tokyo, Japan). The EPR test was carried out on SPINSCAN X (ADANI Company, Minsk, Belarus). The N_2_ physisorption was performed on a JW-BK (JWGB, Beijing, China) surface area and pore size analyzer at the temperature of liquid nitrogen (77 K). The specific surface area was calculated according to the Brunauer-Emmett-Teller (BET) theory while the pore size distribution was assessed by applying the Barrett-Joyner-Halenda (BJH) model to the isotherm adsorption branch.

### 2.6. Photocatalytic Activity

The photocatalytic activity of the as-prepared composites was evaluated in the degradation of CIP using a 500 W Xenon lamp with a UV-cutoff filter (λ > 420 nm). Prior to each experiment, 50 mg catalyst were dispersed in 50 mL of CIP solution (20 mg/L). Before irradiation, the suspension was stirred for 60 min in the dark to reach the adsorption-desorption equilibrium. Then, the reaction was initiated by turning on the irradiation. Samples of 5 mL were taken off every 30 min and centrifuged to remove the photocatalyst before analysis by UV-vis spectrophotometry. The concentration of CIP was determined by monitoring the change in absorbance at 273 nm.

## 3. Results and Discussion

### 3.1. Physicochemical Properties

The crystalline structure of the as-prepared samples was analyzed by XRD. As shown in Figure 1a, the diffractogram of kaolin displays diffraction peaks at 24.98°, 38.54° and 62.46°, which were assigned to the (002), (202) and (060) planes, respectively, in line with the reference card of the standard pattern (JCPDS No.14-0164). Two distinct diffraction peaks at 12.74° and 27.60° in the diffractogram of pure g-C_3_N_4_ corresponded to the (002) and (100) crystal planes of g-C_3_N_4_, according the standard card of JCPDS No. 87-1526. The diffraction peaks of pure CeO_2_ were distinctly observed in the corresponding XRD pattern (JCPDS No.43-1002). The XRD pattern of kaolin/CeO_2_/g-C_3_N_4_ composite displayed the characteristic peaks of g-C_3_N_4_ and CeO_2_ at 27.64° and at 28.4°, respectively, while the characteristic peaks of kaolin were not visible due to the loss of structural ordering and dehydroxylation during the calcination. In addition, compared with the kaolin/CeO_2_/g-C_3_N_4_ sample, the (111) diffraction peaks of CeO_2_ and the (110) diffraction peaks of g-C_3_N_4_ are slightly shifted (Figure 1b), which indicates a lattice distortion in the kaolin/CeO_2_/g-C_3_N_4_ composite [49].

XPS measurements were performed to investigate the elemental composition and chemical states on the surface of as-prepared samples. The survey XPS spectrum of the kaolin/CeO_2_/g-C_3_N_4_ nanocomposite is depicted in Figure 2a and confirms the existence of Ce, O, C, N, Al and Si elements. As shown in Figure 2b, the Ce 3d high resolution XPS spectrum displays peaks at 898.2 and 916.5 eV, which are ascribed to Ce(IV) 3d_5/2_ and Ce(IV) 3d_3/2_, respectively, whereas peaks at 882.4 and 888.8 eV are attributed to Ce(III) 3d_5/2_, and peaks at 900.8 and 907.3 eV are ascribed to Ce(III) 3d_3/2_. In Figure 2c, the signal of O 1s was fitted with two peaks centered at 529.3 and 532.4 eV, corresponding to Ce-O bond and the absorbed oxygen on the CeO_2_ surface, respectively. Figure 2d shows the spectrum of C 1 s, which displays two peaks at 284.8 and 288.2 eV attributed to C-NH_2_ and N-C=N groups, respectively, in g-C_3_N_4_. As shown in Figure 2e, depicting the N 1 s high resolution XPS spectrum, there are three distinct peaks at 398.6, 399.6 and 401.0 eV, which were ascribed to sp^2^ –bonded nitrogen (N-C=N), tertiary nitrogen (N-(C)_3_) and N-H side groups, respectively. The peak at 404.4 eV might be caused by the charge effect. Theoretically, Si and Al elements may originate from the kaolin. Hence, the XPS results confirmed the successful coupling of kaolin and CeO_2_ with g-C_3_N_4_.

EPR was used to investigate the structural defects in kaolin/CeO_2_/g-C_3_N_4_ composite and mono component samples. As shown in Figure 2f, no signal was detected in CeO_2_, suggesting the absence of defects in CeO_2_. However, signals were detected for g-C_3_N_4_, kaolin and kaolin/CeO_2_/g-C_3_N_4_ composite. In addition, it was noticed that the signal intensity of kaolin/CeO_2_/g-C_3_N_4_ was higher than that of g-C_3_N_4_, indicating that the kaolin/CeO_2_/g-C_3_N_4_ composite should exhibit improved catalytic activity, Therefore, the presence of g-C_3_N_4_ had a significant influence on the reactive species because of the changed transfer pathway of charges.

SEM and TEM were employed to investigate the morphology and structure of the as-prepared samples. As shown in Figure 3a,b, a 3D “sandwich” structure was formed by the hydrogen ions functionalized process in the kaolin/CeO_2_/g-C_3_N_4_ composite. The CeO_2_ displays a nanorod morphology, the nanorods having diameters of 50–80 nm. The 2D nanosheet morphology of g-C_3_N_4_ with a wrinkled structure was obtained in the thermal polymerization process. Kaolin had a bulk morphology due to the amorphization and aggregation during the thermal treatment. The kaolin/CeO_2_/g-C_3_N_4_ composite was further investigated by HRTEM. As shown in Figure 3c,d, a lattice spacing of 0.31 nm was observed, which corresponds to the (1-1-1) plane of CeO_2_. The lattice spacing of g-C_3_N_4_ and kaolin was not clearly observed due to poor crystallinity. Energy dispersive spectrometer (EDS) and elemental mapping showed that C, N, O, Si and Ce were uniformly distributed at the surface of the heterostructure (Appendix A).

UV-vis diffuse reflectance spectroscopy was used to study the optical properties of g-C_3_N_4_, CeO_2_, CeO_2_/g-C_3_N_4_ and kaolin/CeO_2_/g-C_3_N_4_ samples, and the spectra are illustrated in Figure 4a.

The spectrum of kaolin/CeO_2_/g-C_3_N_4_ composite displays a peak with maximum at 403 nm, which is broader and stronger than those of pure g-C_3_N_4_, CeO_2_ and kaolin_,_ whereas the optical absorption edge is extended to around 430 nm. However, a low light absorption ability within the range of 445 nm was observed for the kaolin due to the light scattering effect. The band gaps of CeO_2_ and g-C_3_N_4_ were calculated based on Equation (1).
*αhν* = *A(hν − E_g_)*^*n*/2^(1)
where *α**,*
*h, ν, A* and *E_g_* are the optical absorption coefficient, Planck constant, photon frequency, a constant and band gap, respectively. *“n”* denotes the type of electronic transitions, i.e., *n* = 1 or 4 shows a direct-allowed or indirect-allowed transition, respectively. Since g-C_3_N_4_ and CeO_2_ were indirect semiconductors, *E_g_* was estimated from the plot of (*αhν*)^1/2^. Hence, band gaps of 2.73, 2.75 and 2.92 eV were calculated for kaolin/CeO_2_/g-C_3_N_4_ composite, g-C_3_N_4_ and CeO_2_, respectively. Subsequently, the band directly responsible for the interface migration of photoinduced charge carriers was estimated by the Mulliken electronegativity using Equations (2) and (3).
E_CB_ = χ_p_ − E^e^ − 0.5E_g_(2)
E_VB_ = E_CB_ + E_g_(3)
where E_CB_ and E_VB_ are the edge potentials of conduction and valence bands, respectively, χ_p_ is the electronegativity of semiconductor, E^e^ is the energy of free electrons on the hydrogen scale (~4.5 eV), and E_g_ is the band gap energy of semiconductor. The valence band potentials (CB) of g-C_3_N_4_ and CeO_2_ are 1.82, and 2.04 eV, respectively. According to Equation (2), the conduction valence bands (VB) of g-C_3_N_4_ and CeO_2_ were estimated at −0.91, and −0.71 eV, respectively. According to Equation (3).

The specific surface area and pore volume of the as-prepared samples were evaluated from the N_2_ adsorption/desorption isotherms. As shown in Figure 5, the physisorption isotherms of all samples belong to type IV, which indicated their mesoporous structure. The values of the surface area, pore volume and pore diameter are listed in Table 1. The specific surface area of the kaolin/CeO_2_/g-C_3_N_4_ composite was 77.608 m^2^g^−1^, which was higher than that of kaolin (12.628 m^2^g^−1^), CeO_2_ (9.770 m^2^g^−1^) and g-C_3_N_4_ (29.936 m^2^g^−1^). The pore volumes of the kaolin/CeO_2_/g-C_3_N_4_ composite, kaolin, CeO_2_ and g-C_3_N_4_ were 0.129, 0.051, 0.033 and 0.079, respectively. The large surface area and pore volume of the kaolin/CeO_2_/g-C_3_N_4_ composite could be attributed to the uniform dispersion of CeO_2_ particles and the unique architecture of the composite, which could provide more adsorption and active sites and thus, an effective increase in photocatalytic activity.

### 3.2. Photocatalytic Activity

The photocatalytic activity of the samples was evaluated in the degradation of ciprofloxacin (CIP) under visible light irradiation. CIP, a typical antibiotic that is colorless in aqueous solution, can be effectively removed by the effect of photosensitization during visible-light-driven photodegradation process. As shown in Figure 6a, the mixture was kept in the dark for 60 min to achieve adsorption-desorption equilibrium between the antibiotic and photocatalyst. The removal efficiency of CIP was approx. 24% after 60 min in the presence of kaolin/CeO_2_/g-C_3_N_4_. After irradiation, CIP was degraded by kaolin/CeO_2_/g-C_3_N_4_. In the case of all samples, the kaolin/CeO_2_/g-C_3_N_4_ composite showed superior photocatalytic activity and 90% degradation efficiency of CIP was achieved after 150 min, while pure CeO_2_ and g-C_3_N_4_ degraded by approx. 42% and 51%, respectively. The enhanced photocatalytic performance was explained by the formation of heterojunctions between different materials and the abundant adsorption and active sites on the surface of layered kaolin, improved visible-light absorption and separation efficiency of photo-induced, electron-hole charges. As shown in Figure 6b, the photocatalytic degradation followed the first-order kinetics. The kinetics equation expressed by ln (C_0_/C) = kt, where C_0_/C is the normalized concentration of CIP, it is the reaction time, and k is the reaction rate constant (min^−1^), shows a linear relationship between ln (C_0_/C) and time. The results indicated that the rate constant of the reaction over kaolin/CeO_2_/g-C_3_N_4_ composite was of 0.01022 min^−1^, which was about 2.36 and 2.85 folds higher than those for pure CeO_2_ and g-C_3_N_4_, respectively. Overall, the construction of a heterojunction structure decorated on the layered kaolinite with enhanced visible light absorption ability and effective separation of carriers through chemical bond connection, resulted in an enhanced photoactivity activity of g-C_3_N_4_. The stability and reusability experiments were also applied to prove the potential of the ternary composite for practical applications. Figure 6c depicts the time-dependent absorption spectra of CIP degradation over the as-synthesized composite. Changes in UV-vis spectra of CIP in liquid phase over kaolin, CeO_2_, g-C_3_N_4_, kaolin/CeO_2_, kaolin/g-C_3_N_4_ and CeO_2_/g-C_3_N_4_ composite vs irradiation time are shown in Appendix A. A dramatic decrease in the characteristic absorption band of CIP at 273 nm could be observed. As shown in Figure 6d, the kaolin/CeO_2_/g-C_3_N_4_ composite exhibited excellent catalytic performance after four reaction cycles, demonstrating its recyclability and structural stability. The possible mechanism of the photocatalytic degradation of CIP over g-C_3_N_4_/TiO_2_/bentonite nanocomposite was proposed, as illustrated in Figure 7 [50,51,52].

### 3.3. Photocatalytic Mechanism

The electrochemical impedance spectroscopy (EIS) and transient photocurrent responses were carried out in order to further elaborate the photogenerated charge separation and electron transfer performance. As shown in Figure 8a, the smallest arc radius in the EIS Nyquist plot was obtained for the kaolin/CeO_2_/g-C_3_N_4_ composite, indicating a lower interfacial charge-transfer resistance and more effective separation and transfer of photo-generated electron−hole pairs at the interfaces created in the composite. Furthermore, the intensity of photocurrent produced in the kaolin/CeO_2_/g-C_3_N_4_ composite was obviously greater than those of kaolin, CeO_2_ and g-C_3_N_4_ (Figure 8b). Therefore, the highest charge separation efficiency occurred in the g-C_3_N_4_/CeO_2_/kaolin composite.

The radical trapping experiments were conducted to analyze the mechanism of photodegradation process based on the generated radicals. Isopropanol (IPA), disodium ethylenediaminetetraacetate (EDTA-2Na) and 1,4-benzoquinone (BQ) were employed as scavengers of hydroxyl radicals (•OH), holes (h^+^) and superoxide radicals (•O_2_^−^), respectively. As shown in Figure 9a, a slight change in photocatalytic efficiency was observed when IPA was used, suggesting that •OH was not the main active species in the photocatalytic degradation. The photocatalytic efficiency was significantly decreased in the presence of BQ, indicating that •O_2_^−^ intensively influenced the photocatalytic reaction. The photocatalytic activity was evidently reduced when EDTA-2Na was used, indicating that the h^+^ also played a key role in CIP degradation. Hence, h^+^ and •O_2_^−^ were the two main active species in the photocatalytic degradation of CIP solution.

DMPO spin-trapping EPR spectra were used to observe the active radicals involved in CIP photo-degradation process. The signals of DMPO^−^•O_2_^−^ were detected under visible-light irradiation (Figure 9b), confirming that •O_2_^−^ radicals were produced in the photocatalytic reaction. As shown in Figure 9c, a negligible resonance signal of DMPO^−^•OH under darkness and visible light irradiation was observed, suggesting that no or a few •OH radicals participated in the photocatalytic reaction. Hence, the h^+^ and •O_2_^−^ were the principal active species in degradation of CIP.

The experimental results allowed to propose the mechanism of interfacial charge separation and transfer process as well as the possible photocatalytic mechanism involved in the degradation of CIP over a ternary kaolin/CeO_2_/g-C_3_N_4_ heterojunction. Both mechanisms are depicted in Figure 10. Due to the large specific areas and negative-charged characteristic, kaolin promoted an increase in the number of available surface active sites and the separation of the photo-induced carriers [45]. Under visible-light irradiation, CeO_2_ and g-C_3_N_4_ were both excited, and the photogenerated electrons in conduction band (CB) of g-C_3_N_4_ were transferred to CB of CeO_2_, whereas the photogenerated holes in the valence band (VB) of CeO_2_ were transferred to that of g-C_3_N_4_. The photogenerated electrons in CB of CeO_2_ (ECB = −0.71 eV) could reduce O_2_ to •O_2_^−^ (E^0^O_2_/•O_2_^−^ = −0.33 eV vs. NHE) [53]. The accumulated h^+^ in VB of the g-C_3_N_4_ could not directly oxidize the adsorbed H_2_O to produce the •OH, because EVB of the g-C_3_N_4_ was more negative than E^0^•OH/H_2_O (+2.38 eV vs. NHE) [54] as shown in the Table 2. Hence, h^+^ could directly react with the organic pollutant, which was consistent with the results of the radical trapping experiment. In summary, it can be stated that the h^+^ and •O_2_^−^ active species played a significant role in the degradation process.

## 4. Conclusions

In the present study, a novel ternary kaolin/CeO_2_/g-C_3_N_4_ heterojunction was fabricated via a sol-gel method followed by mild hydrothermal treatment. The morphological analysis and other characterizations confirmed the formation of heterojunction, which displayed a higher surface area (77.608 m^2^g^−1^) and the enhanced visible-light absorption. The kaolin/CeO_2_/g-C_3_N_4_ composite exhibited superior photocatalytic activity in CIP degradation under visible light irradiation, and CIP was completely removed within 150 min. The enhanced photocatalytic activity could be attributed to the enhanced light harvesting capability and effective separation of photo-induced electrons and holes. This study provided new insights into the construction of highly efficient ternary photocatalysts under visible-light irradiation with real potential for the organic pollutant degradation.

## Figures and Tables

**Figure 1 materials-13-03811-f001:**
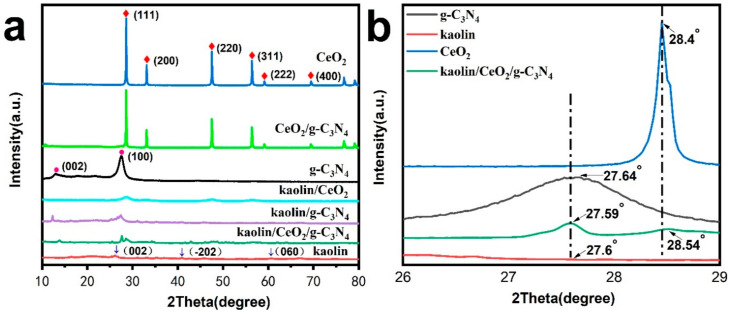
(**a**) XRD patterns of kaolin, CeO_2_, g-C_3_N_4_, kaolin/CeO_2_, kaolin/ g-C_3_N_4_, CeO_2_/g-C_3_N_4_ and kaolin/CeO_2_/g-C_3_N_4_ composite, (**b**) the corresponding enlarged area marked with dashed lines in Figure 1a.

**Figure 2 materials-13-03811-f002:**
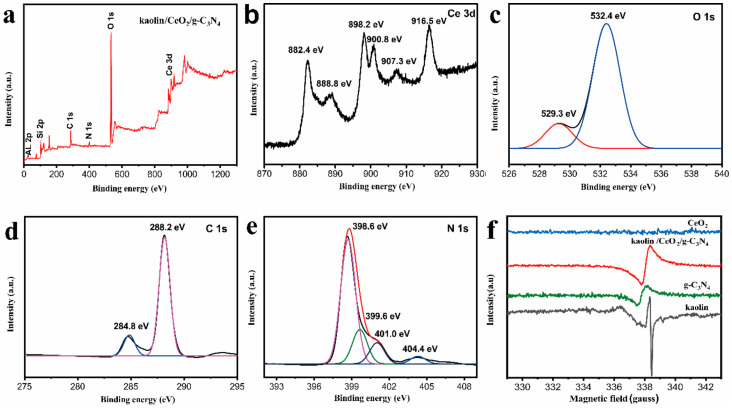
(**a**) XPS spectrum of kaolin/CeO_2_/g-C_3_N_4_ composite, (**b**) Ce 3d, (**c**) O 1s, (**d**) C 1s and (**e**) N 1s (**f**) The EPR signals of CeO_2_, kaolin/CeO_2_/g-C_3_N_4_, g-C_3_N_4_ and kaolin.

**Figure 3 materials-13-03811-f003:**
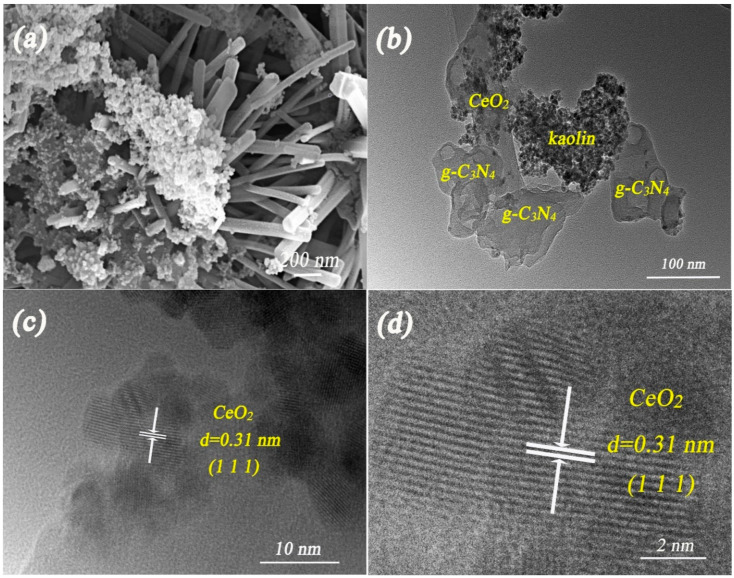
FESEM images of kaolin/CeO_2_/g-C_3_N_4_ composite (**a**); TEM (**b**) and HRTEM images (**c**,**d**) of kaolin/CeO_2_/g-C_3_N_4_ composite.

**Figure 4 materials-13-03811-f004:**
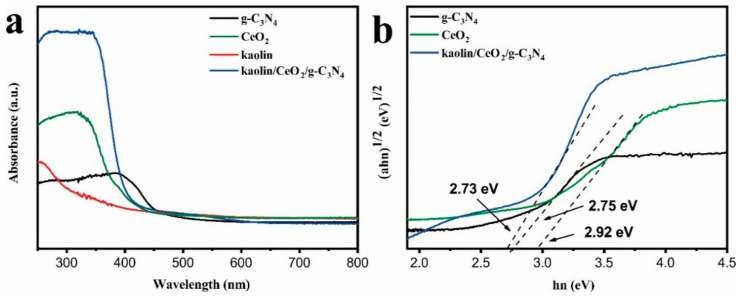
UV-Vis diffuse reflectance spectra (DRS) (**a**) and band gaps (**b**) of kaolin, CeO_2_, g-C_3_N_4_ and the kaolin/CeO_2_/g-C_3_N_4_ composite.

**Figure 5 materials-13-03811-f005:**
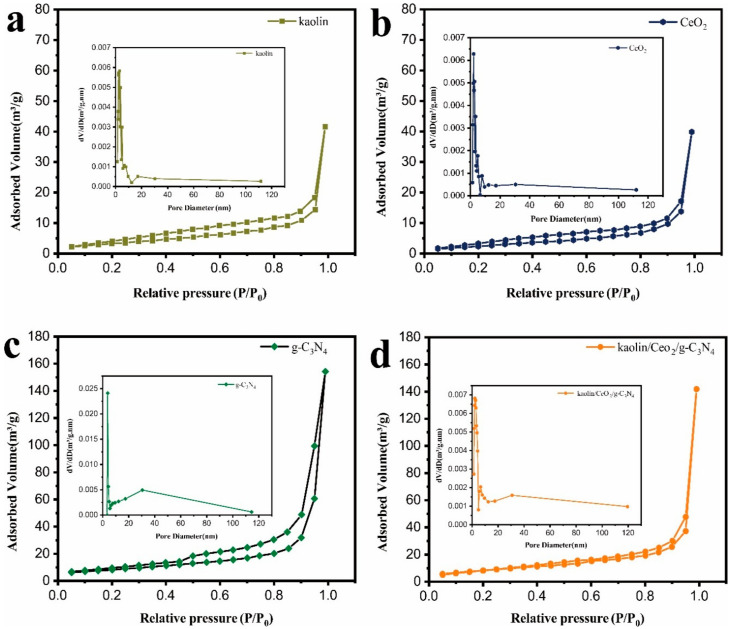
N_2_ adsorption desorption isotherms of (**a**) kaolin, (**b**) CeO_2_, (**c**) g-C_3_N_4_ and (**d**) the kaolin/CeO_2_/g-C_3_N_4_ composite.

**Figure 6 materials-13-03811-f006:**
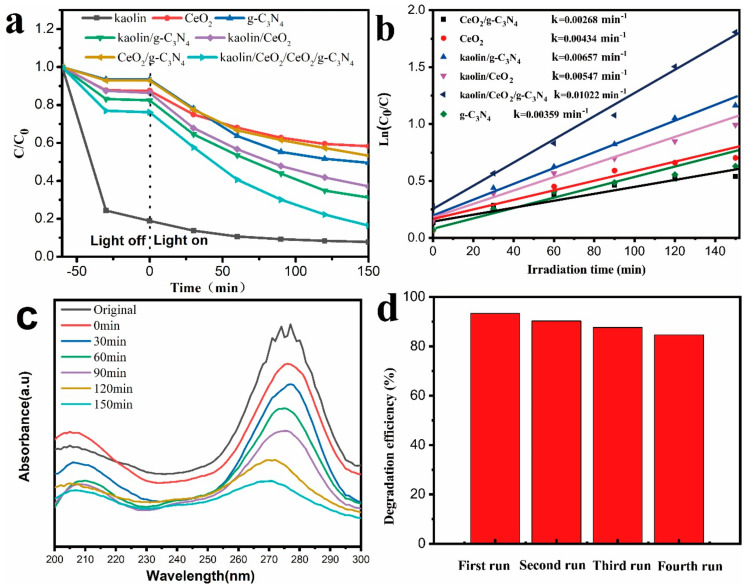
(**a**) Photocatalytic degradation of ciprofloxacin (CIP) under visible light irradiation, (**b**) Linear transform Ln(C0/C) of the kinetic curves of CIP under visible light irradiation, (**c**) The changes in UV-vis spectra of CIP in liquid phase over the kaolin/CeO_2_/g-C_3_N_4_ composite vs irradiation time (**d**) Reusability of the kaolin/CeO_2_/ g-C_3_N_4_ composite.

**Figure 7 materials-13-03811-f007:**
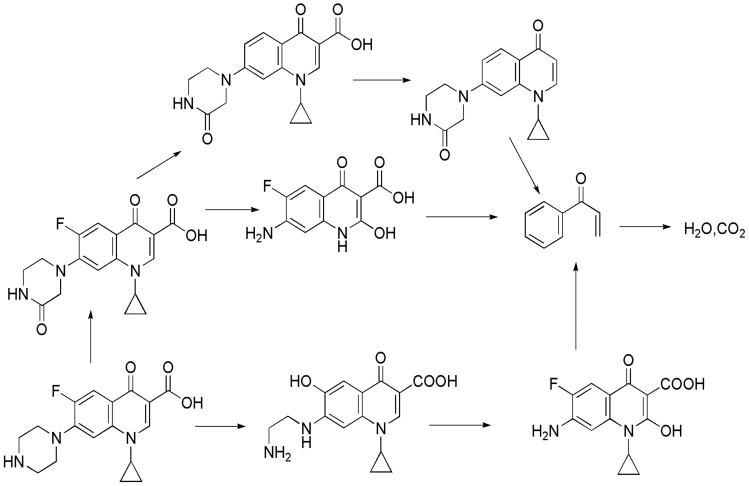
The mechanism of the photocatalytic degradation of CIP over the kaolin/CeO_2_/g-C_3_N_4_ nanocomposite.

**Figure 8 materials-13-03811-f008:**
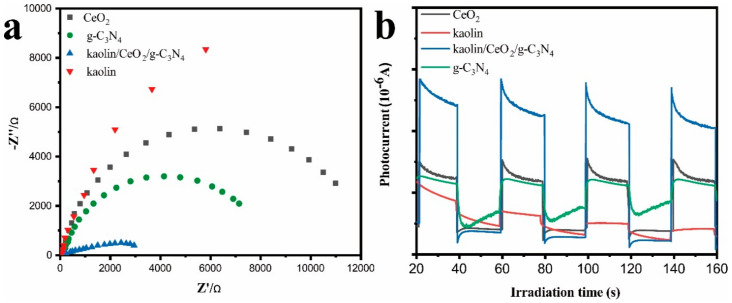
Transient photocurrent and Nyquist plots from electrochemical impedance spectroscopy (EIS) data for the composite and reference samples. (**a**) Nyquist plots for kaolin, CeO_2_, g-C_3_N_4_ and kaolin/CeO_2_/g-C_3_N_4_ composite, (**b**) transient photocurrent produced in kaolin, CeO_2_, g-C_3_N_4_ and kaolin/CeO_2_/g-C_3_N_4_ composite.

**Figure 9 materials-13-03811-f009:**
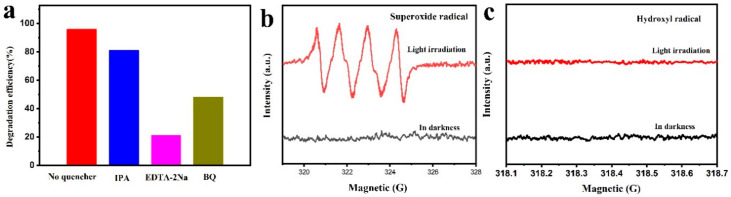
(**a**) Trapping experiments of active species during photodegradation of CIP in the presence of kaolin/CeO_2_/g-C_3_N_4_ under visible light irradiation; (**b**) DMPO^−^•OH and (**c**) DMPO^−^•O_2_^−^ adducts on kaolin/CeO_2_/g-C_3_N_4_ under visible light irradiation.

**Figure 10 materials-13-03811-f010:**
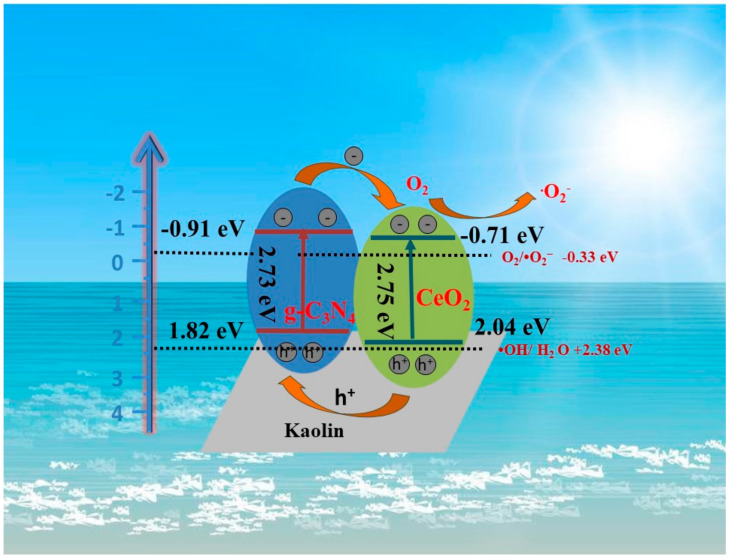
Schematic representation of electron transfer on the surface of the kaolin/CeO_2_/g-C_3_N_4_ heterostructure.

**Table 1 materials-13-03811-t001:** Textural characteristics of the nanocomposites determined from nitrogen sorption measurements.

Sample Name	Slope	Pore Volume (cm^3^/g)	Pore Diameter (nm)	C Constant	Surface Area (m^2^/g)
CeO_2_	341.548	0.0331	8.9552	24.305	9.77
kaolin	268.517	0.0507	13.6668	37.946	12.628
g-C_3_N_4_	115.108	0.0787	19.4854	94.908	29.936
kaolin/CeO_2_/g-C_3_N_4_	44.621	0.1286	31.0499	178.012	77.608

**Table 2 materials-13-03811-t002:** Energy band engineering analysis of g-C_3_N_4_ and CeO_2_.

**Samples**	**E_CB_ for NHE^a^/eV**	**E_VB_ for NHE ^a^/eV**	**E_g_/eV**
g-C_3_N_4_	−0.91	1.82	2.73
CeO_2_	−0.71	2.04	2.75

^a^ NHE: normal hydrogen electrode.

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
