# Peer review of "Synthesis of Novel Kaolin-Supported g-C3N4/CeO2 Composites with Enhanced Photocatalytic Removal of Ciprofloxacin"

_materials, 2020, doi:10.3390/ma13173811_

Round 1

Reviewer 1 Report

While the work appears interesting in the field, the manuscript requires revision before further consideration.

The following comments are appended for the authors:

  1. The authors should present a literature survey on the synthesis methods of kaolin/CeO2, and kaolin/g-C3N4 in the introduction and state their novelty with respect to the synthesis as well.
  2. Section 2.4. Preparation of Kaolin/CeO2 Composite is not clear. While the CeO2  required hydrothermal treatment at 180  ℃  for  24  h for the synthesis, it appears that Kaolin/CeO2  synthesis approach does not use this step. Is that correct? If so, how can the authors compare the performances of the materials since their structure is related to the synthesis conditions?!
  3. In section 2.4. Preparation of Kaolin/CeO 2 Composite it is mentioned ‘CeO 2 /g-C 3 N 4 and  kaolin/g-C3N4  composites were synthesized using the same method’. This is ambiguous, it is recommended to include full description.
  4. Please detail the XPS analysis in the experimental (baseline, deconvolution etc).
  5. Please include the degas/activation/thermal treatment applied to the samples for recording the N2 adsorption isotherm.
  6. No figure is present in the text! Not possible to perform further review.
  7. In the results section: XRD- not clear if authors present also kaolin/CeO2 or kaolin/g-C3N4 composites.
  8. The authors should explain the ‘loss of structural ordering and dihydroxylation during the calcination’ in the XRD results and support with references.
  9. The authors should support the claim ‘the presence of g-C3N4 had a significant influence on the reactive species because of the changed transfer pathway of charges’ for EPR results.
  10. No SEM review can be performed!
  11. The authors should clearly present the estimation: ‘conduction valence bands (VB) of g-C 3 N 4 and CeO 2  were estimated at -0.91, and -0.71 eV’.
  12. Only the ‘uniform dispersion of CeO 2 particles and the unique architecture of the composite’ may not explain the higher specific surface area. Please provide with improved explanation and references.
  13. The authors should also present the photolysis process and provide with adsorption values (as they stated, the composites improves the surface adsorption of the CIP).
  14. No comparison is provided in any of the characterization measurements of the ternary compoasite with the dual ones. In order to better explain the results, such composites should be presented.
  15. No review of the mechanism can be performed due to absence of figures!

Author Response

Journal: Materials

Manuscript ID: Materials-890474

Title: Synthesis of novel kaolin-supported g-C3N4/CeO2 composites with enhanced photocatalytic removal of ciprofloxacin

Author(s): Zhiquan Huang, Leicheng Li *, Zhiping Li, Huan Li, Jiaqi Wu

 Dear Marija Stojic, 

Thank you for your careful evaluation of our submitted manuscript (former manuscript number: Materials-890474). We appreciate the insightful comments from the reviewers. The following letter contains our responses and changes to the reviewer’s comments.

We have revised the manuscript accordingly and answered the reviewers’ questions point by point here. In the light of some comments from the reviewers, some new experimental results have been included to confirm and refine the current results. The language was further improved by the editor of Mogo Editing. The revised parts have been marked in red in the revised version. If you have any further question, please feel free to contact me.

Sincerely yours,

Leicheng Li

North China University of Water Resources and Electric Power

No.36, Beihuan Road, Zhengzhou, Henan, China/Post Code: 450045   

Responds to the reviewer’s comments:

Reviewer 1: Thank you for your suggestions. We are thankful for your letter and for yours’ comments concerning our manuscript. These comments are valuable and very helpful not only to improve our manuscript, but also a source of guidance to our researchers. We have revised the manuscript according to yours’ comments and the detailed revisions can be found in the following responses.

Comment 1: The authors should present a literature survey on the synthesis methods of kaolin/CeO2, and kaolin/g-C3N4 in the introduction and state their novelty with respect to the synthesis as well.

Response: Reviewer’s suggestions have been adopted. We have presented a literature survey on the synthesis methods of kaolin/CeO2 and kaolin/g-C3N4 in the introduction (Appl. Clay Sci. 2016, 129: 7-14). It is well known that kaolin is kind of the most versatile industrial minerals with extensively applications, which can be used for the synthesis of different types of molecular sieves. Using the natural raw Kaolin materials for the production of molecular sieves has obvious economical advantage comparing with the synthetic chemical. In addition, the g-C3N4/CeO2 composite can be modified by adding kaolin to promote the adsorption-desorption property and improve photocatalytic property for CIP degradation. This new type may be promising and widely applied in the oxidation of CIP in the wastewater.

Comment 2: Section 2.4. Preparation of kaolin/CeO2 Composite is not clear. While the CeO2 required hydrothermal treatment at 180 ℃ for 24 h for the synthesis, it appears that kaolin/CeO2 synthesis approach does not use this step. Is that correct? If so, how can the authors compare the performances of the materials since their structure is related to the synthesis conditions?

Response: Thank you very much for your comments. We have supplemented some necessary information in the revised version to make intelligibility of the given information ease. The detailed synthesis process of kaolin/CeO2 was prepared as follows: 1.305 g of Ce(NO3)3•6H2O and 14.400 g of NaOH were dissolved in 60 mL distilled water under magnetic stirring for 2 h. Then, the homogeneous solution was transferred to a 100 mL Teflon-lined stainless steels autoclave and heated at 180 ℃ for 24 h. Subsequently, the autoclave was cooled to room temperature naturally. The resulted precipitate was washed three times with distilled water and ethanol.  Then, the resulting product is added to 60 ml of deionized water and stirred with magnetic force for two hours to form suspended solution A. Kaolin (1.0 g) was immersed in 30 mL of ethanol, magnetically stirred for 30 min and then sonicated for 2 h, marked solution B. Afterwards the suspended solution A was slowly added to the suspension B, stirring for 12 h. The obtained product was dried in an oven at 80 ℃ for 12 h and heated at 450 ℃ (heating rate of 5 ℃/min) for 3 h in a muffle furnace. The obtained sample was kaolin/CeO2.

Corresponding changes in page 3, line 90:

Kaolin/CeO2 composite was prepared as follows: 1.305 g of Ce(NO3)3•6H2O and 14.400 g of NaOH were dissolved in 60 mL distilled water under magnetic stirring for 2 h. Then, the homogeneous solution was transferred to a 100 mL Teflon-lined stainless steels autoclave and heated at 180 ℃ for 24 h. Subsequently, the autoclave was cooled to room temperature naturally. The resulted precipitate was washed three times with distilled water and ethanol.  Then, the resulting product is added to 60 ml of deionized water and stirred with magnetic force for two hours to form suspended solution A. Kaolin (1.0 g) was immersed in 30 mL of ethanol, magnetically stirred for 30 min and then sonicated for 2 h, marked solution B. Afterwards the suspended solution A was slowly added to the suspension B, stirring for 12 h. The obtained product was dried in an oven at 80 ℃ for 12 h and heated at 450 ℃ (heating rate of 5 ℃/min) for 3 h in a muffle furnace. The obtained sample was kaolin/CeO2. For comparison, CeO2 was synthesized by the similar method without kaolin.

Comment 3: In section 2.4 Preparation of kaolin/CeO2Composite it is mentioned ‘CeO2/g-C3N4 and kaolin/g-C3N4 composites were synthesized using the same method’. This is ambiguous, it is recommended to include full description. 

Response: Thank you very much for your comments. The detailed synthesis process of CeO2/g-C3N4 was prepared as follows: 1.305 g of Ce(NO3)3•6H2O and 14.400 g of NaOH were dissolved in 60 mL distilled water under magnetic stirring for 2 h. Then, the homogeneous solution was transferred to a 100 mL Teflon-lined stainless steels autoclave and heated at 180 ℃ for 24 h. Subsequently, the autoclave was cooled to room temperature naturally. The resulted precipitate was collected, washed three times with distilled water and ethanol.  Then, taking a certain the result product is added to 60 ml of deionized water and stirred with magnetic force for two hours to form suspended solution A. Then, a certain amount of g-C3N4 was immersed in 30 mL of ethanol, magnetically stirred for 30 min and then sonicated for 2 h. The resulting suspended solution B. Afterwards the suspended solution A was slowly added to the suspension B, under stirring which was continued for 12 h. The obtained product was dried in an oven at 80 ℃ for 12 h. Then, the solid was ground into powder and heated at 450 ℃ (heating rate of 5 ℃/min) for 3 h in a muffle furnace. The obtained solid was CeO2/g-C3N4.

Kaolin/g-C3N4 composite was prepared as follows: A certain amount of the kaolin was dispersed in an ethanol solution and stirred for 12 h (solution A). A certain amount of g-C3N4 powder was dispersed in an ethanol solution and stirred for at least 8 h (solution B). Then, solution B was added dropwise to the solution A and stirred continuously for 1 h (solution C). The solution C was continuously stirred for 24 h at room temperature. The obtained suspension was centrifuged and the obtained solid dried in an oven at 60 °C for 12 h. The dried sample was ground and then heated to 550 ℃ (heating rate of 5 ℃/min) for 4 h in a muffle furnace. The obtained sample was a kaolin/g-C3N4 composite.

Corresponding changes in Supplementary material data.

Comment 4: Please detail the XPS analysis in the experimental (baseline, deconvolution etc).

Response: Thank you very much for your comments. XPS analysis was performed to investigate the chemical states of the elements in the as-prepared composites. The Peakfit software was used to help us analyze the XPS figure of composites. The peak of C1s was used for correction. In the Ce 3d spectrum (Fig. 2b), the peaks at 898.2 eV and 916.5 eV could be defined to Ce(IV) 3d5/2 and Ce(IV) 3d3/2, while the spectra of Ce(III) 3d5/2 and Ce(III) 3d3/2 were located at 882.4 eV, 888.8 eV, 900.8 eV and 907.3 eV, respectively. As shown in Fig. 2c, the O 1s spectra of CeO2 could be resolved into two peaks centered at 529.3 and 532.4 eV, corresponding to Ce-O bond and the absorbed oxygen on the CeO2 surface, respectively. The Fig. 2d illustrates the spectrum of C 1s, which displays two peaks at 284.8 and 288.2 eV attributed to C-NH2 and N-C=N groups, respectively, in g-C3N4. As shown in Fig. 2e, depicting the N 1s high resolution XPS spectrum, there are three distinct peaks at 398.6, 399.6, and 401.0 eV, which were ascribed to sp2 –bonded nitrogen (N-C=N), tertiary nitrogen (N-(C)3), and N-H side groups, respectively.

Corresponding changes in page 4, line 152.

Comment 5: Please include the degas/activation/thermal treatment applied to the samples for recording the N2 adsorption isotherm.

Response: According to reviewer’s comment, N2 sorption isotherms were measured by a Quantachrome AUTOSORB-1 volumetric gas adsorption analyzer. A liquid N2 bath was used for these measurements. It was used for free space correction measurements. Before each test, about 50-100 mg materials was weighed and degassed at 423 K for 24 h. Then the sample was installed on the Quantachrome AUTOSORB-1 volumetric gas adsorption analyzer.

Comment 6: No figure is present in the text! Not possible to perform further review.

Response: I'm sorry. We uploaded the pictures and manuscript into system separately before. I’m also confused about this mistake. I have uploaded pictures again in this new manuscript.

Comment 7: In the results section: XRD-not clear if authors present also kaolin/CeO2 or kaolin/g-C3N4 composites.

Response: Thank you very much for your suggestions. Kaolin/CeO2 and kaolin/g-C3N4 composites of XRD analysis have been included, as shown in Fig 1a.

Figure 1. (a) XRD patterns of kaolin, CeO2, g-C3N4, CeO2/ g-C3N4, kaolin/CeO2, kaolin/g-C3N4 and the as-prepared kaolin/CeO2/g-C3N4 composite, (b) the corresponding enlarged area marked with dashed lines in Fig.1(a).

Comment 8: The authors should explain the ‘loss of structural ordering and dihydroxylation during the calcination’ in the XRD results and support with references.

Response: Thank you very much for your comments. We’ve recognized that this description is not accurate. The sentence “loss of structural ordering and dihydroxylation during the calcination” is changed to “loss of structural ordering and dehydroxylation during the calcination”. Due to the calcinations, the ordered stacking between the kaolin layers disappeared and the dehydroxylation reaction in the kaolin completed, leading to that the characteristic peaks of kaolin disappeared. The references have been cited in the revised manuscript (Applied Catalysis B: Environmental. 2018, 220: 272-282).

Corresponding changes in page 4, line 144:

The XRD pattern of kaolin/CeO2/g-C3N4 composite displayed the characteristic peaks of g-C3N4 and CeO2 at 27.64° and at 28.4°, respectively, while the characteristic peaks of kaolin were not visible due to the loss of structural ordering and dehydroxylation during the calcination.

Comment 9: The authors should support the claim ‘the presence of g-C3N4 had a significant influence on the reactive specie because of the changed transfer pathway of charges’ for EPR results.                      

Response: As shown in the Fig. 2f, May be the presence of g-C3N4 had a significant influence on the reactive specie because of the changed transfer pathway of charges, so that the trend of paramagnetic wave of composite materials is basically consistent with that of C3N4.

Figure. 2. (a) XPS spectrum of kaolin/CeO2/g-C3N4 composite, (b) Ce 3d, (c) O 1s, (d) C 1s and (e) N 1s (f) The EPR signals of CeO2, kaolin/CeO2/g-C3N4, g-C3N4 and kaolin.

Comment 10: No SEM review can be performed!

Response: Thank you for your suggestion. The results of SEM have been supplemented in the revised manuscript.

Figure 3. FESEM images of kaolin/CeO2/g-C3N4 composite (a); TEM images (b) and HRTEM images (c-d) of kaolin/CeO2/g-C3N4 composite.

Comment 11: The authors should clearly present the estimation: ‘conduction valence bands (VB) of g-C3N4 and CeO2 were estimated at -0.91, and -0.71 eV’.

Response: Thank you very much for your comments. The valence bands (VB) of g-C3N4 and CeO2 were estimated by the Mulliken electronegativity expressed by Eq (1) and (2)

ECBp-Ee-0.5Eg                                                    (1)

EVB=ECB+Eg                                                       (2)

The band gap (Eg) and ECB were calculated, where ECB and EVB are edge potentials of respectively conduction and valence bands, χp is electronegativity of semiconductor, Ee is energy of free electrons on the hydrogen scale (∼4.5 eV), and Eg is band gap energy of semiconductor. According to Eq. (1), the conduction bands (CB) potentials of g-C3N4 and CeO2 were estimated at -0.91, and -0.71 eV, respectively. According to Eq. (2), the valence band (VB) potentials of g-C3N4 and CeO2 are 1.82 and 2.04 eV, respectively.

Comment 12: Only the ‘uniform dispersion of CeO2 particles and the unique architecture of the composite’ may not explain the higher specific surface area. Please provide with improved explanation and references.

Response: Thank you very much for your suggestion. I'm sorry for that the conclusion is a theoretical hypothesis and I have no way to give you a better explanation based on the current experimental data.

Comment 13: The authors should also present the photolysis process and provide with adsorption values (as they stated, the composites improves the surface adsorption of the CIP).

Response: Thank you very much for your suggestion. The Fig. 6(a) and Fig. 6(c) presented the adsorption and photolysis process. Before irradiation, the adsorption values of as-prepared samples were clearly shown on the diagram, and the kaolin/CeO2/g-C3N4 composite possessed the stronger adsorption capacity for the CIP, which was consistent with the result the BET. After irradiation, the kaolin/CeO2/g-C3N4 composite showed the optimum photocatalytic efficiency, and the removal efficiency of ciprofloxacin (CIP) was about 90% within 150 min, 2-folds higher than those of pristine CeO2 and g-C3N4.

Figure 6. (a) Photocatalytic degradation of CIP under visible light irradiation, (b) Linear transform Ln(C0/C) of the kinetic curves of CIP under visible light irradiation, (c) The changes in UV-vis spectra of CIP in liquid phase over kaolin/CeO2/g-C3N4 composite vs irradiation time (d) Reusability of kaolin/CeO2/ g-C3N4 composite.

Comment 14: No comparison is provided in any of the characterization measurements of the ternary composite with the dual ones. In order to better explain the results, such composites should be presented.

Response: We agree with the reviewer comment. With respect to the comparison of the characterization measurements of the ternary composite with the dual ones,it is better to present the dual composites in the present study. The dual composites were not characterized in the current experiment partly because it was clearly evidenced and descripted in the previous document. In the present study, we pay more attention to the characterization measurements of the ternary composite.

Comment 15: No review of the mechanism can be performed due to absence of figures!

Response: We deduced the mechanism of photocatalytic reaction based on the following data:

(1) The valence band and conduction band of semiconductors are calculated by UV-vis and the Mulliken electronegativity expressed by Eq (1) and (2)

ECBp-Ee-0.5Eg                                               (1)

EVB=ECB+Eg                                                  (2)

(2) The trapping and EPR experiments demonstrated that the hole (h+) and superoxide radicals (•O2) played an important role in the photocatalytic process.

It is concluded that the heterojunction is a typical type II heterojunction based on the band structure and the results of trapping experiment.

Reviewer 2 Report

The manuscript entitled "Synthesis of novel kaolin-supported g-C3N4/CeO2 composites with enhanced photocatalytic removal of ciprofloxacin" by Zhiquan Huang, Leicheng Li, Zhiping Li, Huan Li and Jiaqi Wu describes the preparation of a innovative kaolin/CeO2/g-C3N4 composite and its enhanced photocatalytic activity towards ciprofloxacin (CIP) under visible-light irradiation than the pristine g-C3N4. The  advanced methods such as TEM, XRD, XPS, BET, SEM, EIS, EPR, and DRS were used by authors in order to investigate the structure and morphology of composite. Unfortunately, the figures and tables were missing, so a proper review could not be made in this form of manuscript.

Author Response

We highly appreciate your review and comments. I'm very sorry. According to my previous experience in submission, I uploaded the manuscript to the system separately from the picture and the form, which resulted in no picture and the form in the manuscript, this is probably the main reason you didn't see pictures and charts. Therefore, I have inserted profit pictures and tables in the new manuscript.Please see the attachment.

Reviewer 3 Report

Author has investigated on the synthesis of novel kaolin-supported g-C3N4/CeO2 composites with enhanced photocatalytic removal of ciprofloxacin. The experiments have been well-conducted and the manuscript seems to be novel. I would like to suggest few comments to author to consider before its publication.

  1. It would be interesting to see the elemental mapping image of the obtained ternary composite because its difficult to differentiate kaolin and CeO2 in Fig. 3b.
  2. Author can add the UV-Visible absorption curves of CIP over the other catalyst (Fig. 6c), which would be useful for readers.

Round 2

Reviewer 1 Report

The authors improved the quality of their manuscript by responding to the review comments. However, comment 13 still need to be addressed as no photolysis process is shown in the figures and minor English editing is advised (e.g. L90-101).

Author Response

Responds to the reviewer’s comments:

Comment 1: The authors improved the quality of their manuscript by responding to the review comments. However, comment 13 still need to be addressed as no photolysis process is shown in the figures and minor English editing is advised (e.g. L90-101).

Response: Thanks for the reviewer's kind suggestion. In order to clearly show the photocatalytic degradation process, we made a slight adjustment to the graph of photocatalytic degradation. As shown in Fig. 6a, prior to irradiation, the removal efficiency of CIP was about 24% within 60 min in the presence of kaolin/CeO2/g-C3N4, which could be caused by adsorption. During this period, we believed that the adsorption-desorption equilibrium had been reached. After irradiation, the photocatalytic degradation was the main way to remove the CIP, which could further decompose to smaller intermediate products or be mineralized to CO2 and H2O. The removal efficiency of CIP was 90% within 150 min under visible light irradiation. Except for the adsorption, the extra part was considered as photolysis process (Fig. 6a).

Figure 6. (a) Photocatalytic degradation of CIP under visible light irradiation.

Corresponding changes in page 8, line 247-253:

As shown in Fig. 6a, the mixture was kept in the dark for 60 min to achieve adsorption-desorption equilibrium between the antibiotic and photocatalyst. The removal efficiency of CIP was approx. 24% after 60 min in the presence of kaolin/CeO2/g-C3N4. After irradiation, CIP was degraded by kaolin/CeO2/g-C3N4. In the case of all samples, the kaolin/CeO2/g-C3N4 composite showed superior photocatalytic activity and 90% degradation efficiency of CIP was achieved after 150 min, while pure CeO2 and g-C3N4 degraded by approx. 42% and 51%, respectively.

In the light of some comments from the reviewers, a native-English speaker helped us to refine our manuscript carefully. Changes in the revised manuscript are highlighted in red in our new submission.

Corresponding changes in page 3, line L90-101:

Ce(NO3)3•6H2O (1.305 g) and NaOH (14.400 g) dissolved in distilled water (60 mL) under magnetic stirring for 2 h. Then, the homogeneous solution was transferred to a 100 mL Teflon-lined stainless steels autoclave and heated at 180 ℃ for 24 h. Subsequently, the autoclave was naturally cooled to room temperature. The resulting precipitate was washed three times with distilled water and ethanol. The product was added to deionized water (60 mL) and magnetically stirred for 2 h producing suspended solution A. Kaolin (1.0 g) was immersed in ethanol (30 mL), magnetically stirred for 30 min and sonicated for 2 h producing solution B. Then, suspended solution A was slowly added to suspension B with stirring for 12 h. The obtained product kaolin/CeO2 was dried in an oven at 80 ℃ for 12 h and heated at 450 ℃ (heating rate of 5 ℃/min) for 3 h in a muffle furnace. For comparison, CeO2 was synthesized using a similar method without kaolin.

This manuscript is a resubmission of an earlier submission. The following is a list of the peer review reports and author responses from that submission.